# Cadmium Sulfide Quantum Dots, Mitochondrial Function and Environmental Stress: A Mechanistic Reconstruction through In Vivo Cellular Approaches in *Saccharomyces cerevisiae*

**DOI:** 10.3390/nano13131944

**Published:** 2023-06-26

**Authors:** Marta Marmiroli, Giovanni Birarda, Valentina Gallo, Marco Villani, Andrea Zappettini, Lisa Vaccari, Nelson Marmiroli, Luca Pagano

**Affiliations:** 1Department of Chemistry, Life Sciences and Environmental Sustainability, University of Parma, 43124 Parma, Italy; 2Elettra, Sincrotrone Trieste, Strada Statale 14—km 163.5 in AREA Science Park, Basovizza, 34149 Trieste, Italy; 3Istituto dei Materiali per l’Elettronica e il Magnetismo, Consiglio Nazionale delle Ricerche (IMEM-CNR), 43124 Parma, Italy; 4Consorzio Interuniversitario Nazionale per le Scienze Ambientali (CINSA), University of Parma, 43124 Parma, Italy

**Keywords:** mitochondrial function, *Saccharomyces cerevisiae*, mechanisms, cadmium sulfide quantum dots, Fourier-transform infrared spectroscopy (FTIR), deletion mutants

## Abstract

Research on the effects of engineered nanomaterials (ENMs) on mitochondria, which represent one of the main actors in cell function, highlighted effects on ROS production, gametogenesis and organellar genome replication. Specifically, the mitochondrial effects of cadmium sulfide quantum dots (CdS QDs) exposure can be observed through the variation in enzymatic kinetics at the level of the respiratory chain and also by analyzing modifications of reagent and products in term of the bonds created and disrupted during the reactions through Fourier-transform infrared spectroscopy (FTIR). This study investigated both in intact cells and in isolated mitochondria to observe the response to CdS QDs treatment at the level of electron transport chain in the wild-type yeast *Saccharomyces cerevisiae* and in the deletion mutant *Δtom5*, whose function is implicated in nucleo–mitochondrial protein trafficking. The changes observed in wild type and *Δtom5* strains in terms of an increase or decrease in enzymatic activity (ranging between 1 and 2 folds) also differed according to the genetic background of the strains and the respiratory chain functionality during the CdS QDs treatment performed. Results were confirmed by FTIR, where a clear difference between the QD effects in the wild type and in the mutant strain, *Δtom5*, was observed. The utilization of these genetic and biochemical approaches is instrumental to clarify the mitochondrial mechanisms implicated in response to these types of ENMs and to the stress response that follows the exposure.

## 1. Introduction

Budding yeast (*Saccharomyces cerevisiae*) has proven to be a functional model organism to study a variety of basic cellular functions conserved in higher eukaryotes. Several features make yeast particularly suitable for biochemical, genetic and cell biology studies [1]. For example, yeast is capable of satisfying its energy requirements with adenosine triphosphate (ATP) generated when growing on organic carbon sources using either mitochondrial respiration or glycolysis [1]. During evolution, most of the genetic functions originally belonging to the endosymbiotic prokaryote DNA were transferred into the nucleus. These include a few factors essential for iron/sulfur cluster assembly, the import and assembly of nuclear-encoded precursor proteins and flavin mononucleotide synthesis in the mitochondria. The fact that many nuclear-encoded mitochondrial functions can be studied using knock-out nuclear mutants makes yeast an ideal organism for exploring the molecular processes required for the biogenesis of respiratory-competent mitochondria [2]. Moreover, it is becoming increasingly clear how cytosolic and mitochondrial proteostasis are intimately related. These processes can play a role in shaping cell fate [3]. Yeast is one of the leading candidates to replace animals in toxicology testing of new generation materials because many of the cellular mechanisms associated with cytotoxicity, adaptation and tolerance to chemical and environmental stress are preserved from yeast to higher eukaryotes [4,5]. Among the numerous types of engineered nanomaterials (ENMs) whose utilization has been assessed in terms of utility as well as risks associated with environmental safety and human health in recent years, quantum dots (QDs) are one of the most frequently considered ENMs [6,7,8,9,10,11,12]. QDs applications include imaging detection systems and diagnostic tools for biomarkers of cancer cells [13], immune assays, cancer therapy [14] or vehicles for DNA, protein and drug delivery to pathogenic cells [15].

The information gained from studying *S. cerevisiae*’s response to QDs at the physiological and molecular levels highlighted the importance of the mitochondrion as one of the main targets involved in the environmental stress response [16]. Quantum dots, because of their physico-chemical properties mainly enter living cells by endocytosis [17], thus determining specific cellular and genetic responses [5,7,16].

However, more direct evidence is needed to construct a more mechanistic picture of the QD–yeast interaction. One of the responses to QD exposure is related to intracellular Reactive Oxygen Species (ROS) production directly derived from QD interaction with mitochondrial membranes at the level of the electron transport chain, leading to oxidative stress induction, mitochondrial function reduction and mitochondrial morphology disruption and apoptosis [5,7,9,16].

Fourier-transform infrared spectroscopy (FTIR) has a wide application range, from small molecules to molecular complexes to analyses of cells or tissues. The tissue and cell imaging had recent developments in infrared spectroscopy, taking advantage of the use of synchrotron IR radiation and of infrared microscopy. It is utilized for mapping the cellular components (carbohydrates, lipids, proteins) and to identify the abnormal or degenerated cells [18,19]. FTIR spectroscopy is increasingly applied to study proteins, with regard to folding, conformation, and molecular details of protein active sites during enzymatic reactions, by using reaction induced difference spectroscopy (FTIR) [20]. In specific, infrared spectroscopy probes molecular vibrations. Characteristic infrared absorption bands can be associated with functional groups, corresponding to fundamental vibrations of functional groups [21]. When N atoms are in a nonlinear molecule, the vibrational motions are 3N-6, which corresponds to normal modes. A normal mode of vibration that absorbs the incident infrared light is infrared active when there is a change in the molecule dipole moment. Symmetric vibrations are generally not detected in the infrared range. In particular, when molecules have a center of symmetry, all vibrations that are symmetrical with respect to the center are infrared inactive. Contrarily, molecular asymmetric vibrations are detected. The lack of selectivity allows probing properties of a large number of chemical groups in a single sample, with particular regard provided to water molecules and amino acids, which can prove difficult when observing with other spectroscopic techniques [22].

In this work, FTIR spectroscopy was applied to study the changes at the level of respiratory complexes of the mitochondria in intact cells or isolated mitochondria from *S. cerevisiae* treated with CdS QDs. Results were compared with the classical enzymatic assays to demonstrate the applicability of this technique to the study of mitochondrial phosphorylation complexes in vivo, their fate and function during the response as well as during adaptation to environmental stresses. As an applied biology proof of concept, several targets from respiratory chains (complexes II, III, IV) were considered to assess the organellar functionality under stress conditions (CdS QDs treatment) or in the absence of some mitochondrial function through gene deletion in a specific QD target, such as Tom5, as demonstrated by Marmiroli et al., 2016 [5]. Interestingly, by comparing the CdS QDs and Cd ion exposure, a marked overlap between commonly isolated sensitive mutants is not observed [5,23]; only 11 mutants on a panel of 114 isolated as CdS QD sensitive mutants were commonly observed as sensitive to Cd^2+^ exposure (9.6%), highlighting the differential response mechanisms exploited. Particular emphasis has been given in the case of CdS QDs response to abiotic stress and mitochondrial organization, while Cd^2+^ toxicity response was mainly related to metal detoxification and vacuole transport [5,23]. Differences in the CdS QD and Cd^2+^ responses can be also testified by the differential MIC observed for CdS QDs (250 mg L^−1^) and CdSO_4_ (50 μM, which corresponds to 56 mg L^−1^ of Cd^2+^) [23].

The *Saccharomyces cerevisiae* strain BY4742 (MATα), which harbors the mutations his3Δ1, leu2Δ0, lys2Δ0 and ura3Δ0 and the isogenic knock-out mutant (EUROSCARF) ypr133w-a (*Δtom5*), were utilized for FTIR and biochemical assays. The *TOM5* gene encodes for a component of the TOM (translocase of outer membrane) complex, which is responsible for the initial import of the mitochondrially directed proteins [24]. Tom5 plays a role in maintaining the TOM complex structural integrity, and it is essential for the tail-anchored components interaction of the TOM complex with Tom40, the pore-forming protein [24,25]. The potential impairments at the level of protein trafficking from the endoplasmic reticulum to the mitochondrion may influence different functions of the organelle itself, including the respiratory chain [23]. A detailed characterization of the effects of CdS QDs from the proteomic point of view utilizing different separation methods, such as 2D PAGE and iTRAQ, has been carried out, confirming the pivotal role of mitochondrion as the main target of the CdS QD exposure [26].

## 2. Methods

### 2.1. Experimental Setup

The strain *ypr133w-a* (*Δtom5*) was identified as sensitive by mutant collection screening performed by Marmiroli et al. (2016) [5] in the condition of treatment with CdS QDs, while *Δtom5*’s molecular and functional characterization (ROS production, glutathione redox state, oxygen consumption, abundance of respiratory cytochromes, mitochondrial morphology) in the condition of CdS QD exposure were fully elucidated by Pasquali et al., 2017 [7]. The strains were grown in a YPD (2% *w*/*v* Dextrose, 1% *w*/*v* yeast extract, 2% *w*/*v* Peptone) liquid medium with 0 mg L^−1^ or with 100 mg L^−1^ CdS QDs [7] and using the sonicated Fisher Scientific Model 505 Sonic Dismembrator (Fisher Scientific, Waltham, MA, USA), set at a 40% amplitude for 60–120 s to maximize dispersion. Exposure tests have been conducted using sublethal conditions: a prior estimation of the CdS QDs minimal inhibitory concentration (MIC) was carried out (250 mg L^−1^) using concentrations ranging from 0 to 250 mg L^−1^ [5,7]. All reagents were purchased from Merck (Darmstadt, Germany).

### 2.2. Cadmium Sulfide Quantum Dots Characterization

Cadmium sulfide quantum dots (CdS QDs; size: <5 nm; purity: 99.99%; composition: 78% Cd and 22% S) were synthesized by IMEM-CNR (Parma, Italy). QDs were fully characterized, as reported in Pagano et al. (2022) [27]; the QDs’ average particle size (178.7 nm) and zeta (ζ) potential (+15.8 mV) were determined in ddH_2_O by Zetasizer Nano Series ZS90 (Malvern Instruments, Malvern, UK). QD dissolution in ddH_2_O, for 10 days, has been investigated by ultracentrifugation and atomic absorption spectroscopy (FA-AAS; AA240FS, Agilent Technologies, Santa Clara, CA, USA). Dissolution rate was estimated as 0.5%.

### 2.3. Mitochondrial Extraction

Measurements were performed using intact cells or isolated mitochondria. Mitochondria were isolated from CdS QD treated and untreated cells using the mechanical lysis process described by Nedeva et al., 2002 [28], with several modifications: In specific, 2.5 × 10^8^ cells mL^−1^ were resuspended in 1.2 M sorbitol and 20 mM phosphate buffer supplemented with 1 mM protease inhibitor cocktail (P2714). Mechanical lysis was performed in 1.5 mL Eppendorf tubes with glass beads (425–600 μm) by grinding for 10 s at an oscillation frequency of 3 Hz for 3 times at 4 °C. Cells debris was precipitated using 3000 rpm, 5 min, 4 °C centrifugation, and the supernatant were recovered. A second centrifugation step at 10,000 rpm, 10 min and 4 °C allowed the precipitation of mitochondria with resuspension in fresh 1.2 M sorbitol and 20 mM phosphate buffer supplemented with 7% *v*/*v* BSA buffer (75 mg mL^−1^ BSA, 0.6 M sorbitol). All reagents were purchased from Merck (Darmstadt, DE, USA).

### 2.4. Fourier-Transform Infrared Spectroscopy (FTIR) Measurements

FTIR measurements were performed at the Elettra Sincrotrone Trieste facility accessed through the CERIC-ERIC proposal 20177035. The following benchtop equipment of SISSI-Bio beamline [29] was used: Vertex 70 interferometer (Bruker Optics, Billerica, MA, USA), deuterated triglycine sulfate (DTGS) detector (Infrared Laboratories Inc., Tucson, AZ, USA), and MIRacle™ single reflection ATR (attenuated total reflection) (Pike Technologies, Madison, WI, USA) equipped with a diamond crystal and thermalized at 37 °C using an in-house made thermal jack, which is also used to prevent evaporation of the medium. Analyses were performed on intact cells (5 × 10^4^ cells) or on isolated mitochondria (20 μg) following Barrientos et al. (2009) [30], with a reduction of the final volume of the reaction from 1 mL to 100 µL. We performed each assay in 3 replicates after the deposition of a 10 µL drop of sample suspension in medium onto the attenuated total reflectance (ATR) crystal and the addition of the specific substrate for each step of the respiratory chain. A series of repeated spectra were collected averaging 64 scans with a spectral resolution of 2 cm^−1^ (3 min) for every time point for 30 min. A reference spectrum was collected before each run on the cleaned crystal. After the acquisition, the data were corrected for atmospheric contribution (water vapor and CO_2_) using the OPUS 7.5 software (Bruker Optics, Billerica, MA, USA). Then, the data were imported in QUASAR (https://quasar.codes, accessed on 1 June 2022), interpolated to 2 cm^−1^, smoothed using the Savitzky Golay filter (window 13 points, polynomial degree 2, derivative order 0), cut in the 1480–800 cm^−1^ range, baseline corrected and vector normalized. After the pre-processing, the data were analyzed using principal components analysis (PCA) and the results were plotted using Origin (Pro) Version 2022 (OriginLab Corporation, Northampton, MA, USA). Two strains of *S. cerevisiae* were analyzed: a wild-type strain (wt) and the mutant *Δtom5*, which were hypersensitive to the QD exposure [5,7]. FTIR measurements were performed by analyzing the response of the cells or the isolated mitochondria upon the activation of specific complexes of the respiratory chain: complex II (Succinate Decylubiquinone DCPIP reductase, SQR), complex III (Succinate Cytochrome c reductase, SCCR) and complex IV (Cytochrome c oxidase).

### 2.5. Enzymatic Assays

Measurements on intact cells (5 × 10^4^ cells) or on isolated mitochondria (20 μg) were performed by analyzing the activity of different enzymatic functions involved in the mitochondrial electron transport chain with regard to complex II, complex III and complex IV activities. The enzymatic activity of each complex was analyzed separately following the specific enzymatic assay described in detail in Barrientos et al. (2009) [30]. The measurement of succinate decylubiquinone DCPIP reductase (SQR) (complex II) activity was performed as follows: In 1 mL of SQR medium, mitochondria or whole cells, 4 μM rotenone, 80 μM DCPIP as acceptor and 0.2 mM ATP were added. Additionally, 10 mM succinate was utilized as the donor. Incubation was performed for 10 min at 30 °C. The reaction started by adding 80 μM decylubiquinone. The activity resulting from the reduction of 2,6-dichlorophenolindophenol (DCPIP) was measured for 5 min at 595 nm. Furthermore, the measurement of succinate cytochrome c reductase (SCCR) (complex III) activity was conducted as follows: In 1 mL of SCCR medium, mitochondria or whole cells, 4 μM rotenone, 240 μM KCN and 0.2 mM ATP were added, and 10 mM succinate was utilized as the electron donor again. Subsequently, incubation for 10 min at 30 °C was performed. The reaction started by adding 40 μM oxidized cytochrome c, and activity was measured for 5 min at 550 nm. Similarly, the measurement of cytochrome c oxidase (complex IV) activity was performed as follows: In 1 mL of isosmotic COX medium, mitochondria, or whole cells, 10 μM reduced cytochrome c and 2.5 mM lauryl maltoside (to permeabilize the mitochondrial outer membrane) were added. The absorbance at 550 nm was measured for 3 min. Analyses were carried out by the Varian Cary 219 UV–VIS spectrophotometer (Agilent Technologies, Santa Clara, CA, USA). A two-tail Student’s *t*-test (*p* < 0.01) was performed on intact cells or isolated mitochondria using IBM SPSS, v.24.0 (Chicago, IL, USA; http://ibm.com/analytics/us/en/technology/spss/, accessed on 1 June 2022).

## 3. Results and Discussion

### 3.1. FTIR Analyses

Principal component analysis (PCA, Figure 1 and Table 1) elaborates the results obtained from the analyses for complexes II, III and IV, respectively. The principal components in terms of the total variance in the systems were selected among those that allowed for good separation between the two strains, did not contain strong signals from the buffer and allowed the replicates to cluster together, i.e., are highly affected by differences coming from the testing environment. For all the experiments, there is a clear separation between the datasets from the wt and *Δtom5*: complexes II and III showed a similar separation between wt and *Δtom5* mutant along abscissae. Moreover, complex II data from treatments with QDs separate from controls along PC3, suggesting that the treatment with QDs may affect both wt and *Δtom5* strains in a similar way. Data acquired after the activation of complex IV (Figure 1c) show that the treatment with QDs more effectively impacted *Δtom5*, which separates from the others along PC3.

Regarding complex III, from the infrared signals, it is possible to hypothesize that exposure to QDs did not alter the vibrational features and signal since these properties overlap between treated cells and untreated controls. Figure 1d–f showed the loading vectors depicting the signals that are related to the separation of the measurement points in clusters. From their analysis, it is possible to appreciate that the vector associated with the separation between the two strains shows similar features in all experiments. In fact, PC2 for complex II and IV and PC1 for complex III are characterized by the same main peaks, as can be seen from the black dotted lines in Figure 1d–f at wavelengths of 1655 cm^−1^, 1545 cm^−1^ (that overlaps with a red line) and 1413 cm^−1^. The first two peaks can be related to amide I and II and variations in the protein conformation between the two strains, derived by the different states of protonation of the aminoacidic residues implicated in the electron transport [31], while the third can be assigned to the ring C-C, CH_2_ and CH_3_ bonds, which may imply a potential dysfunction in the electron transport chain [7]. Hellwig et al. (2000) [32] stated that the region below 1200 cm^−1^ may be affected by the signals from buffer, whose phosphate groups are influenced by the H^+^ exchanges during the reaction; therefore, strong signals in PC1 or PC2 will be considered coming from the buffer. Instead, PC3, the ordinate axis for all three scatterplots, represents, at best, 2% of the variance of the system. It is reasonable to assign this signal to the subtle rearrangements occurring during the different steps of the respiratory processes, which proceeds differently in wt and *Δtom5*. From the literature, similar processes can be observed in amide I/II changes that could be associated with conformational changes related to the protonmotive mechanism [33], which are consistent with the reactions performed during the assay [30]. Within this range, in Complex II, two negative peaks at 1690 and 1621 cm^−1^ were observed, both of which could be assigned to beta structures, and one positive at 1657 cm^−1^, linked to alpha helix structures. At lower wavelengths, the signals from quinone/quinol coupling at 1465, 1443 and 1385 cm^−1^ were observed [34]. In the PC3 for complex III, again, it is possible to observe a broad positive band centered at 1695 cm^−1^, a sharp peak at 1652 cm^−1^ and another negative peak at 1543 cm^−1^, all referable to amide I and amide II polypeptide backbone changes. According to other reports, the peak at 1695 cm^−1^ can be associated with a variation of the iron–sulfur protein (ISP) of the cytochrome bc_1_ complex [35,36,37,38]. The positive signals from 1496 to 1381 cm^−1^ can be assigned to the quinol [39]. Lastly, in Figure 1f, the red line shows PC3 for the experiments with complex IV. In this case, there were fewer strong signals indicating protein rearrangements detected from amide I variations, but a positive peak at 1665 cm^−1^ and a broad negative one at 1625 cm^−1^ are present and are related to the peaks at 1546 and 1516 cm^−1^ in amide II, respectively [33]. Through FTIR analysis, therefore, it is possible to distinguish the response of wt and *Δtom5* cells and identify the main rearrangements occurring during the treatments even though the experiment is not time resolved per se.

Measures conducted on intact cells showed that the signatures of all the other proteins and organelles can mask the spectral features of the process itself. Therefore, similar experiments were carried out on isolated mitochondria from the same yeast cells, and the results are shown in Figure 2 (and summarized in Table 1). PCA data showed that, for both complexes (II and IV), the separation between conditions (treated vs. untreated) was larger than between strains. In fact, in Figure 2a,b, by observing the data along PC1, we find that the controls notably cluster close to each other, and the same happens for QDs-treated mitochondria. For the complex II activation experiment, this separation is less clear and in both, untreated and treated wt, mitochondria are partially overlapping. For the *Δtom5* strain, the controls are separated from QDs treatment, but this later overlaps with the treated wt strain. The PC1 spectral features related to this dataset are shown in Figure 2c (black line). From left to right, the main peaks are 1695 cm^−1^, 1616 cm^−1^, 1464 and 1413 cm^−1^, the first two of which are assignable to proteins’ rearrangements involving beta sheets conformations, whereas the second two can be assigned to carboxylic acids, such as succinate and fumarate, involved in the reaction [40]. PC5 in Figure 2c (red line), instead, accounts for the small variations within the experiments. It can be noticed that QDs-treated mitochondria slightly lean toward the upper left quadrant of the scatterplot when compared with the respective controls. Therefore, it can be hypothesized that the signals related to this component are correlated to a different response of the organelles in the presence of QDs. Data regarding the activation of complex IV show that strong signals can be observed in the range of proteins at 1657 and 1611 cm^−1^. Similarly, in Figure 2a, data of the untreated controls cluster close along PC1 in the right hemi-plane, and the QDs-treated mitochondria cluster in the left hemi-plane, with the latter spread more along PC4.

**Figure 2 nanomaterials-13-01944-f002:**
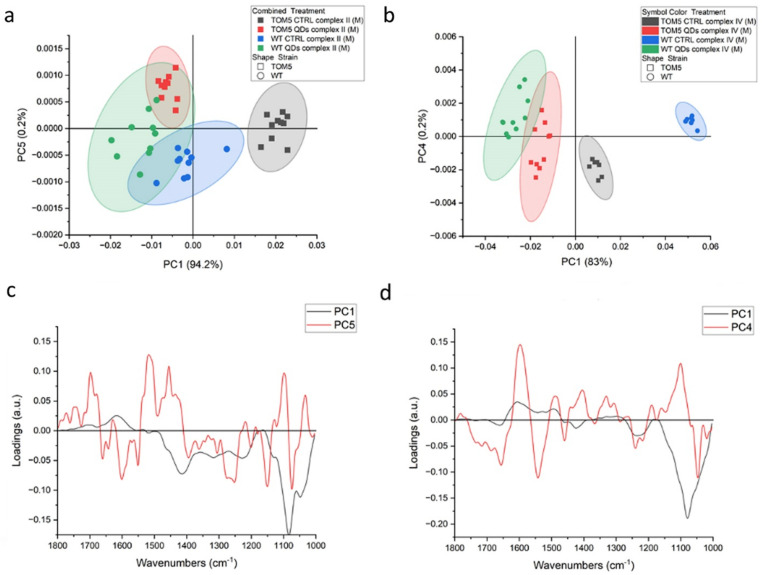
PCA analysis of FTIR spectra from isolated mitochondria of *Saccharomyces cerevisiae* after the activation of respiratory chain complexes II (**a**) and IV (**b**). Loading vectors representing PC1 and PC5 (**c**) and loading vectors representing PC1 and PC4 (**d**) are both related to complex II analyses (**a**). Major components indicated are related to the treatment performed on the two yeast strains (untreated vs. CdS QD treatment), reported in green and blue (wt) or black and red (mutant), respectively.

### 3.2. Colorimetric Assays and Comparison with FTIR

The biochemical assays confirmed the trend of the observation obtained with FTIR analyses: differences and commonalities between the two sets of experiments. Data from biochemical assays, performed in cells or in isolated mitochondria, are reported in Figure 3 and Table 1. Specifically, analyzing complex II activity (Figure 3a) showed a reduction, particularly when comparing the *Δtom5* strain in treated and untreated conditions. The change in activity led to results similar to that observed in the wt strain treated with QDs. Furthermore, the analysis of complex III showed a strong effect related to the treatment (Figure 3b). In fact, both treated wt and *Δtom5* strains resulted in a decrease in activity between 30s and 60s, followed by a strong increase in the activity at 300 s. The results are comparable to those obtained in the FTIR analyses (Figure 1a,b) since the effects of the QD treatment seem to be more remarkable on *Δtom5* as compared to the wt under the same conditions. Figure 3c showed that the untreated wt and mutant strains seem to be quite similar in their activity, while, when treated with CdS QDs, a strong decrease in complex IV activity in *Δtom5* was observed, which showed a constitutive reduced activation from the reaction’s start.

In isolated mitochondria (Figure 3d), no significant changes in complex II activity were observed. A partial correspondence can also be observed with the enzymatic assay performed on the whole cell. The untreated *Δtom5* strain only showed differences compared to the assay performed on isolated mitochondria (Figure 3a). The results are similar to those observed with FTIR analyses on isolated mitochondria (Figure 2a). Figure 3b,e showed only a partial correspondence between intact cell and isolated mitochondrial results, highlighting how in Complex III a decrease in activity is observed in the case of the wt strain treated with QDs, while the *Δtom5* strain did not show significant changes when compared to the untreated strain. Analyses performed on complex IV in isolated mitochondria (Figure 3f) resembled those obtained with the analyses performed on intact cells (Figure 3c): the activity of complex IV in both strains showed a significant decrease in the condition of treatment. The results are compatible with those observed with FTIR on isolated mitochondria (Figure 2b). However, mitochondria may have been affected by the extraction, which can moderately alter the functionality of the complex due to the mechanical lysis protocol, explaining the variability observed in both experiments.

**Figure 3 nanomaterials-13-01944-f003:**
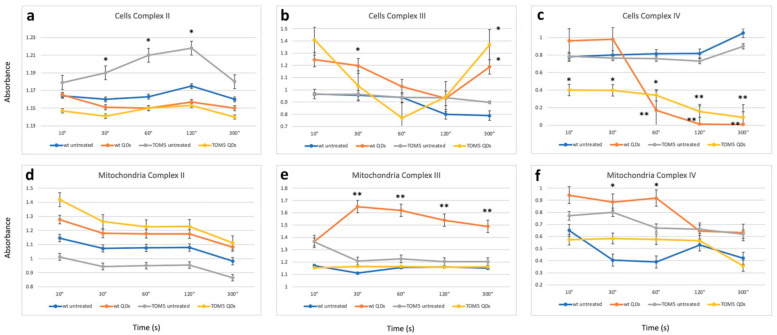
Enzymatic assays performed after the activation of respiratory chain complex II (**a**), complex III (**b**) and complex IV (**c**) in intact cells and isolated mitochondria (**d**–**f**). Time (in seconds) and normalized absorbance are reported on the x-axis and y-axis, respectively. Student’s *t*-test has been applied to the differences observed by comparing the wild-type (wt) untreated samples with the other treatments. Activation kinetics represent wild type untreated (blue), wild type treated with CdS QDs (brown), *Δtom5* strain untreated (gray) and *Δtom5* mutant treated with CdS QDs (yellow). *t*-test: *, *p* < 0.05; **, *p* < 0.01. Analyses performed both on cell and isolated mitochondria suggest that the differences observed (ranging between 1 and 2 folds) can be mainly correlated with the treatment performed (untreated vs. CdS QD treatment).

A cross-comparison of the two experiments highlighted potential concurrence or non-concurrence of the obtained data between FTIR as a direct measurement of the modification in the molecular structure of the components implicated in the reactions, and the biochemical assay as an indirect measurement, which revealed colorimetric variations due to the modification in the structure of a reagent triggered by the mitochondrial environment. Therefore, the two methods can be considered complementary. The major aspects observed are correlated with (i) the genetic background of the two *S. cerevisiae* strains, which are completely isogenic, with the exception of *TOM5* gene deletion, and (ii) the CdS QDs treatment performed. This is consistent with the fact that, with the exception of the treatment and the genetic background of the strains utilized, all the other possible variables (medium, growth temperature and any other environmental conditions) were maintained constant throughout the experiments.

These two parameters can be extrapolated from data analysis (reported in Figure 1, Figure 2 and Figure 3 and summarized in Table 1), highlighting a different sensitivity between the two techniques used to investigate the physiological response in intact cells and isolated mitochondria. The response of the isolated mitochondria can be related to direct impairment in mitochondrial functionality that CdS QD exerts, as previously shown: a reduction in the content of respiratory cytochromes and in the oxygen consumption reduces the capacity of the cells to grow on non-fermentable carbon sources [7].

The results highlighted the differences between the wild type and the knockout mutant, *Δtom5*, whose limited protein trafficking between nucleus, cytoplasm and mitochondrion leads to a decreased crosstalk, which mainly regulates the response to extracellular and intracellular changes but is enough to maintain the cellular homeostasis as well [3,41]. At the transcriptomic and proteomic level, *Tom*5 was identified as a key factor involved in the mitochondrial organization, essential in the stabilization of the TOM complex and favoring the nucleo–mitochondrial protein trafficking [5,25,26,42].

The utilization of CdS QDs as environmental stressors to stimulate the cellular response was purposeful since the CdS QDs’ mechanisms of translocation into the cell could be identified, which include corona-protein modifications and/or trojan horse vesicle transportation [43,44]. After uptake, CdS QDs can be biotransformed within the cellular environment, interacting with different organelles and subcellular targets, including the nucleus and the mitochondria [45], prompting a high reactive oxygen species (ROS) production. It was shown that, in yeast, after 24 h of treatment with CdS QDs, proteins belonging to the mitochondrial respiration complexes III, IV and V were strongly down-regulated [26]. In this context, the mitochondrion, which plays a pivotal role in ROS production, is also involved in the oxidative stress response and in programmed cell death regulation [3,9]. The utilization of constitutively defective mitochondria, as those present in the *Δtom5* mutant strain, was beneficial in studying the effects and mechanisms developed by the yeast cell to deal with the interaction with CdS QDs, as fully characterized by using the genetic point of view through knock-out mutant screening [5], the physiological point of view and, in terms of mitochondrial organization [7], by using the functional point of view at the transcriptomic and the proteomic level [26,42].

## 4. Conclusions

The utilization of the yeast species *S. cerevisiae* as a model system showed two main facilities: (i) the possibility to grow yeast under controlled conditions and (ii) the flexibility of its life cycle, which can occur on fermentable or non-fermentable carbon sources. This study offered the unique opportunity to study the response in mutants whose mitochondrial functionality was compromised [4,5,7]. Knock-out mutants in specific mitochondrial functions, such as *TOM5*, are instrumental to test the effects of artificial stressors, such as ENMs, when some of their potential targets are absent and to compare the results with the behavior of the isogenic wild-type strain [5]. The cellular response to environmental stress (the treatment), which is mainly driven by the mitochondrial response at different levels, consists of the following processes: DNA replication, protein synthesis and assembly, protein trafficking, respiration and oxidative phosphorylation [3,45]. The absence of a complete functional network of proteins in the mitochondrial membrane (as the site of the respiratory chain) allowed for a more clear-cut explanation of the functions involved (in the effects), which goes well beyond the production of ROS and the consequent oxidative stress.

The evidence obtained is relevant from three perspectives: (i) From the methodological point of view, by comparing differential techniques, in order to magnify specific changes in biomolecular structure, organelle functionality and cell viability; (ii) from the genetic point of view, by comparing the behavior of differential yeast strains (wild type vs. isogenic deletion mutant in term of nucleo–mitochondrial protein trafficking); and (iii) by CdS QDs treatment, as a major stressor applied in the field of biology to test the response to stress. It is important to remind, as described in detail by Marmiroli et al., 2016 [5], that CdS QDs and Cd^2+^ exploit different response mechanisms, highlighted by the very low overlap between genes and functions involved in the two pathways [5,42]. Moreover, the differential response mechanisms that interested Cd in nano and ionic forms, fully characterized at physico-chemical level by Pagano et al., 2022 [27], have been deeply elucidated in different model systems, from simple eukaryotes to plants to human cells [5,27,46,47].

In the panel of potential methodologies which can be applied to the comprehension of in vivo mechanisms of homeostasis and cellular response, FTIR is a relatively new entry [48,49]. This technique can be successfully utilized to study, in detail, the variation in the biomolecular structure of reagents and products during any biochemical reaction. The possibility of successfully applying this approach to in vivo studies is instrumental in understanding the biological interactions and relative interactors implicated in the basic molecular pathways investigated. On the other hand, considering nanomaterials, as any other potential stressor, FTIR can be applied to the functional (direct or indirect) investigation of the physico-chemical forms present in the cells, organs or tissues, with particular regard to ENM in vivo biotransformation as an essential step to understand the differences between the particle behavior when compared to their ionic counterparts in complex systems (e.g., living organisms), as was already proved in other eukaryotic model systems [45,50]. Moreover, biochemical assays, due to robustness and complementarity, cannot be merely shelved. Their utilization with Fourier-transform infrared spectroscopy can reinforce enzyme kinetics. The results obtained on mitochondrial respiration and energy production [5,7,16,42,47], together with the data on CdS QDs response, are important to consider since the functional dysregulations observed may have long range effects, such as modification of mitochondrial DNA replication [27], deregulation of mitochondrial proteins [26] and gametogenesis [17], suggesting how mitochondria in *S. cerevisiae* can be considered as the center of epigenetic effects [51,52] during response to environmental stress.

## Figures and Tables

**Figure 1 nanomaterials-13-01944-f001:**
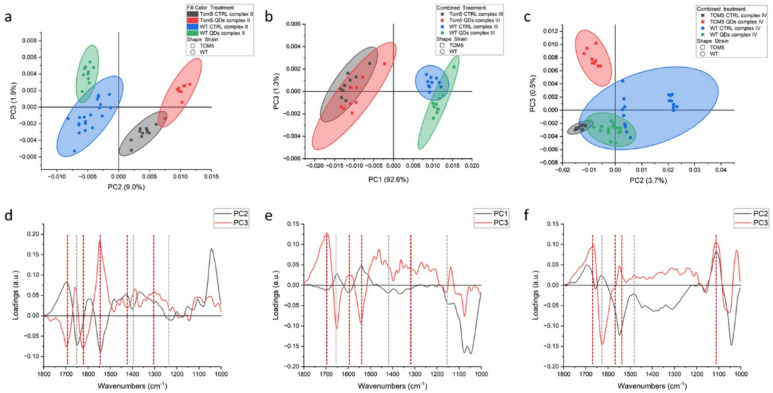
PCA analysis of FTIR spectra from *Saccharomyces cerevisiae* intact cells after the activation of respiratory chain complex II (**a**), complex III (**b**) and complex IV(**c**). Loading vectors (**d**) representing PC2 and PC3 in dotted lines, related to complex II analyses (**a**), indicate the main peaks for PC2 (black) and PC3 (red), respectively. Loading vectors representing PC1 and PC3 (**e**) in dotted lines, related to complex III analyses (**b**), indicate the main peaks for PC1 (black) and PC3 (red), respectively, and loading vectors (**f**) representing PC2 and PC3 in dotted lines, related to complex IV analyses (**c**), indi-cate the main peaks for PC1 (black) and PC3 (red), respectively. Major components from PCA are related to the differential genetic background of the two yeast strains used, reported in green and blue (wt) or black and red (*Δtom5*), for treated and untreated conditions, respectively.

**Table 1 nanomaterials-13-01944-t001:** Summary of the compared results obtained by FTIR and enzymatic analyses (Figure 1, Figure 2 and Figure 3).

**FTIR**	**Intact Cells**	**Isolated Mitochondria**
Complex II	Major differences due to genetic background	Major differences due to effects of the treatment
Complex III	Major differences due to genetic background	-
Complex IV	Major differences due to genetic background	Major differences due to effects of the treatment
**Enzymatic assay**	**Intact cells**	**Isolated mitochondria**
Complex II	Major differences due to genetic background	Major differences due to effects of the treatment
Complex III	Major differences due to effects of the treatment	Major differences due to effects of the treatment
Complex IV	Major differences due to effects of the treatment	Major differences due to effects of the treatment

## Data Availability

Not applicable.

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
