# Peer review of "Cadmium Sulfide Quantum Dots, Mitochondrial Function and Environmental Stress: A Mechanistic Reconstruction through In Vivo Cellular Approaches in Saccharomyces cerevisiae"

_nanomaterials, 2023, doi:10.3390/nano13131944_

Round 1

Reviewer 1 Report

Marmiroli et al have done an interesting study on the effect of cadmium sulfide quantum dots on the function of several electron transport complexes in yeast which are cultured with these dots.  These authors are the leading scientists in this field having published numerous papers on the effects of these dots on mitochondrial of various cell types.  In this paper, they use two techniques to assess mitochondrial effects as well as using whole cells versus isolated mitochondria. In this way they are able to tease out cytosolic/nuclear effects from mitochondria effects. This paper is well written and I only had minor issues with it, listed below.

Minor Comments:

1.     Put Table 1 _after_ Figure 3 where data is presented for the colorometric assays, because it is more of a summary of results.

2.     Figure 3 graphs need y-axis labels, even it if it is for fold change.  I still am unable to understand the x-axis degree numbers, so perhaps a better explanation in the text or figure legend would help folks like me (I know it has something to do with the dots, but still am confused)

One thing that was not explicitly addressed in the discussion is what happens if the cytosolic components (maybe iron containing proteins that aren’t in mitochondria) interfere with the assay results and block them instead of merely having the whole cell data represent the effects of the cytosol/nucleus.  To me, this means that the isolated mitochondria results are the true results and the whole cell results contain artifacts and probably should not be stressed in the overall conclusions.  Looking at the abstract, however, the emphasis is put on the effects of TOM protein, where it seems the conclusion is the emphasis on the CdS dots on mitochondrial function.  Please clarify the abstract or explain why I am mistaken here.  I have done lots of these colorometric studies usually in mitochondria isolated from tissue and always used isolated mitochondria because otherwise, they just don’t work properly, so I don’t see why this is any different in yeast, but I would like to hear your explanation

English is fine.  Just a few minor tense issues.

Author Response

Marmiroli et al have done an interesting study on the effect of cadmium sulfide quantum dots on the function of several electron transport complexes in yeast which are cultured with these dots.  These authors are the leading scientists in this field having published numerous papers on the effects of these dots on mitochondrial of various cell types.  In this paper, they use two techniques to assess mitochondrial effects as well as using whole cells versus isolated mitochondria. In this way they are able to tease out cytosolic/nuclear effects from mitochondria effects. This paper is well written and I only had minor issues with it, listed below.

We thank the Rev1 for the consideration.

Minor Comments:

  1. Put Table 1 _after_ Figure 3 where data is presented for the colorometric assays, because it is more of a summary of results.

We agree on this point. Table 1 has been moved after Figure 3.

  1. Figure 3 graphs need y-axis labels, even it if it is for fold change. I still am unable to understand the x-axis degree numbers, so perhaps a better explanation in the text or figure legend would help folks like me (I know it has something to do with the dots, but still am confused)

We thank Rev 1 for the comment. In order to improve the understanding of the Figure 3, the x-axis and y-axis were named on the figure, and the caption has been also modified accordingly:

“Enzymatic assays performed after the activation of respiratory chain complex II (a), complex III (b) and complex IV(c) in intact cells and isolated mitochondria (d, e, f). Time (in seconds) and normalized absorbance are reported respectively on x-axis and y-axis. Student’ t-test has been applied on differences observed by comparing the wild type (wt) untreated samples with the other treatments. Activation kinetics represented wild type untreated (blue), wild type treated with CdS QDs (brown), Dtom5 strain untreated (gray) and Dtom5 mutant treated with CdS QDs (yellow).”

One thing that was not explicitly addressed in the discussion is what happens if the cytosolic components (maybe iron containing proteins that aren’t in mitochondria) interfere with the assay results and block them instead of merely having the whole cell data represent the effects of the cytosol/nucleus. To me, this means that the isolated mitochondria results are the true results and the whole cell results contain artifacts and probably should not be stressed in the overall conclusions. Looking at the abstract, however, the emphasis is put on the effects of TOM protein, where it seems the conclusion is the emphasis on the CdS dots on mitochondrial function.  Please clarify the abstract or explain why I am mistaken here.  I have done lots of these colorometric studies usually in mitochondria isolated from tissue and always used isolated mitochondria because otherwise, they just don’t work properly, so I don’t see why this is any different in yeast, but I would like to hear your explanation.

We agree with Rev1 that the most important results are related to the isolated mitochondria, although, the utilization of the intact cell has been included as a methodological comparison to understand differences and commonalities of effects determined by the CdS QD treatment on yeast cell, which represent the background situation in which CdS QD response has been assessed from different perspectives: genomic (KO mutant screening), transcriptomic (microarray) and proteomic (iTRAQ vs 2D electrophoresis), as reported in previous works (refs. 5,7,25,41). In this context the Table 1 is functional to highlight those differences, as reported in the discussion, in particular focusing the attention “why” and “how” those differences can be appreciated, both technically and biologically, that is one of the main goal of our analyses.

In this context, we have to remind how the present manuscript represent an applied biology “proof of concept” to offer alternative strategies to observe the biological phenomena and to understand the importance of the complementarity between the observations from the classical biochemical approach (represented in this case by the colorimetric assay) and the more innovative FTIR (bio)physical approach. For this reason, the utilization of a well characterized knock-out mutant, such as TOM5, becomes a fundamental tool in highlighting the differential responses at the mitochondrion functional level between the wild type strain and the mutated (delta)tom5 in case of: i) CdS QDs treatment or ii) with intact cells or isolated mitochondria.

For these reasons, in the abstract we decided to give relevance to TOM5 as tool to describe the effects of the CdS QD exposure on an already known target such as the mitochondria.

Reviewer 2 Report

The manuscript entitled “Cadmium sulfide quantum dots, mitochondrial function and environmental stress: a mechanistic reconstruction through in vivo cellular approaches in Saccharomyces cerevisiae” is significant in this field of interest. The manuscript is well-structured with enough data. This manuscript has a few issues needed to be addressed. Thus I recommend this manuscript for minor revision.

Please review the manuscript for any typographical errors.

Eg. 2.5 × 108 cells mL-1 were

 Please ensure that the statistical software packages used in this study are included in the methods section.

 In Figure 3, the titles for the Y and X axes are missing. Please add them.

 Could you please double-check the statistical analysis (ANOVA) for Figure 3 and confirm its accuracy?

 The discussion should be enhanced with more recent references.

 The conclusion should be derived from the current research findings and should highlight the significance of the study. Please rewrite the conclusion accordingly.

Minor editing of English language required

Author Response

The manuscript entitled “Cadmium sulfide quantum dots, mitochondrial function and environmental stress: a mechanistic reconstruction through in vivo cellular approaches in Saccharomyces cerevisiae” is significant in this field of interest. The manuscript is well-structured with enough data. This manuscript has a few issues needed to be addressed. Thus I recommend this manuscript for minor revision.

We thank the Rev2 for the consideration.

Please review the manuscript for any typographical errors.

Eg. 2.5 × 108 cells mL-1 were

Typos have been checked and corrected throughout the manuscript.

Please ensure that the statistical software packages used in this study are included in the methods section.

Information required has been included.

Lines 206-208: “A two-tail Student’s t-test (p < 0.01) was performed on intact cells or isolated mitochondria using IBM SPSS v. 24.0 (Chicago, Illinois, USA; http://ibm.com/analytics/us/en/technology/spss/).”

In Figure 3, the titles for the Y and X axes are missing. Please add them.

Figure 3 and relative caption were modified according to Rev2 comment:

“Enzymatic assays performed after the activation of respiratory chain complex II (a), complex III (b) and complex IV(c) in intact cells and isolated mitochondria (d, e, f). Time (in seconds) and normalized absorbance are reported respectively on x-axis and y-axis. Student’ t-test has been applied on differences observed by comparing the wild type (wt) untreated samples. Activation kinetics represented wild type untreated (blue), wild type treated with CdS QDs (brown), Dtom5 strain untreated (gray) and Dtom5 mutant treated with CdS QDs (yellow).”

Could you please double-check the statistical analysis (ANOVA) for Figure 3 and confirm its accuracy?

We thank Rev2 suggestion. Data has been checked multiple times. A two-tail Student’s t-test (p < 0.01) has been considered a robust and reliable test for the dataset utilized.

The discussion should be enhanced with more recent references.

It is important to consider that, at the present state, very few publications on the utilization of FTIR as technique to characterize cellular, but in particular mitochondrial response there are differential stressors.

Our proposal is to offer a multidisciplinary and wide open approach to explain a biological phenomenon, that in this case has been consider in very specific context, as a proof of concept i) to introduce a potential, novel and reliable biophysical assay, which may offer a new perspective to analyze phenomena already described from biochemical and genetical point of view; ii) to add novel background information to the already studied pathways and mechanisms implicated in the nucleo-mitochondrial protein trafficking and how this impact with mitochondrial functionality. For this reason the already characterize deletion mutant (delta)tom5, defective in import of mitochondrial proteins, has been taken in to account; and to give a further perspective to study the effect of a potential “next-gen” environmental stressor such as CdS QDs, which have been deeply studied in our lab, on different eukaryotic models (Marmiroli et al., 2016; Pasquali et al., 2017; Gallo, et al, 2021; Pagano et al., 2019; refs 5,7,25,41), in which all the information about mitochondrial target, transcriptional and expression levels have been fully characterized. Additional references have been included in discussion and conclusions.

Discussion has been modified as reported:

Lines 364-412: “A cross-comparison of the two experiments highlighted potential concurrence or non-concurrence of the obtained data between FTIR, as direct measurement of the modification in molecular structure of the components implicated in the reactions, and the biochemical assay as an indirect measurement, which described at colorimetric variation due to the modification in the structure of a reagent triggered by mitochondrial environment: the two methods can be considered therefore complementary. The major aspects observed are correlated with i) the genetic background of the two S. cerevisiae strains, which are completely isogenic, with the exception of TOM5 gene deletion, and ii) with the CdS QDs treatment performed. This is consistent with the fact that, with the exception of treatment and genetic background of the strains utilized, all the other possible variables (medium, growth temperature, any other environmental condition) were maintained constant throughout the experiments.

[…]

The utilization of constitutively defective mitochondria, as those present in the (delta)tom5 mutant were functional to study effects, and mechanism developed by the yeast cell to deal with the interaction with CdS QDs, as fully characterized on genetic point of view through knock-out mutant screening [5], physiological point of view, and in term of mitochondrial organization [7], on functional point of view, at transcriptomic and proteomic level [25,41].”

The conclusion should be derived from the current research findings and should highlight the significance of the study. Please rewrite the conclusion accordingly.

We thank Rev 2 for the suggestions. Conclusions have been modified accordingly:

Lines 416-455: “Utilization of yeast S. cerevisiae as a model system showed two main facilities: i) the possibility to grow yeast under controlled conditions, and ii) the flexibility of its life cycle which can occur on fermentable or non-fermentable carbon sources. This offers the unique opportunity to study the response in mutants whose mitochondrial functionality was compromised [4,5,7]. Knock-out mutants in specific mitochondrial function, as TOM5, are instrumental to test effects of artificial stressors, such as ENMs, when some of their potential targets are absent and to compare the results with the behaviour of the isogenic wild type strain [5]. The cellular response of an environmental stress (the treatment), which mainly is driven by the mitochondrial response at different levels: DNA replication, protein synthesis and assembly, protein trafficking, respiration, and oxidative phosphorylation [3,45]. The absence of a complete functional network of proteins in the mitochondrial membrane (as site of the respiratory chain) allowed for a more clear-cut of functional involved (in the effects) which goes well behind the production of ROS and the consequent oxidative stress.

Evidence obtained is relevant from three perspectives: i) from the methodological point of view, by comparing differential techniques, in order to magnify specific changes in biomolecules structure, organelle functionality, and cell viability; ii) from the genetic point of view, by comparing the behavior of differential yeast strains (wild type vs iso-genic deletion mutant), and iii) by CdS QDs treatment, as a major applied biology proof of concept of response to stress. It is important to state, as described in detail from Marmiroli at al., 2016 [5], that CdS QDs and Cd2+ exploit different mechanisms of response, highlighted by the very low overlap between genes and functions involved in the two pathways [5,41].

In the panel of potential methodologies which can be applied to the comprehension of in vivo mechanisms of homeostasis and cellular response, FTIR is a relatively new entry [46]. This technique can be successfully utilized to study in details the variation in the biomolecular structure of reagents and products during any biochemical reaction. The possibility to successfully apply this approach to in vivo study is instrumental to the understanding of the biological interactions, and relative interactors implicated in basic molecular pathway investigated. On the other hand, considering nanomaterials, as any other potential stressor, FTIR can be applied to the functional (direct or indirect) investigation of the physico-chemical forms present into the cells, organs, or tissues, with particular regard to ENM in vivo biotransformation, as was already proved in other eukaryotic model systems [44,47]. Biochemical assays, due to robustness and complementarity, cannot be merely shelved. Their utilization with Fourier-transform infrared spectroscopy can reinforce the enzymatic kinetics. The results obtained on mitochondrial respiration, energy production [5,7,16,41,48], together with the data on CdS QDs response are important to consider since the functional dysregulations observed may have long range effects such as, modification of mitochondrial DNA replication [26], deregulation of mitochondrial proteins [25], and gametogenesis [17], suggesting how mitochondria in S. cerevisiae can be considered as center of epigenetic effects [49,50] during response to environmental stress.”

Reviewer 3 Report

Authors in their article entitled: Cadmium sulfide quantum dots, mitochondrial function and environmental stress: a mechanistic reconstruction through in vivo cellular approaches in Saccharomyces cerevisiae put their effort in my opinion into the CdS application in living organisms. The idea to use nanomaterials toward diagnostic or therapeutic treatment is important from a social point of view. Moreover, the ENMs application in biological experiments looks attractive too. Due to the above authors should strongly express the possibility of ENMs application in the introduction part. Moreover, in the whole article, I did not find any information about LD50 of CdS and toxicity of CdS in the experiment’s conditions. The toxicity of cadmium should be discussed and presented. Additionally, if authors discussed the mitochondria as a target it is obligated to show the level of discussed proteins (ELIS, WB, ect.). It is well known that mitochondria recruit proteins from the cytosol. Therefore, even though the article is well written and readable I cannot recommend it for publication in its current form.

Author Response

Authors in their article entitled: Cadmium sulfide quantum dots, mitochondrial function and environmental stress: a mechanistic reconstruction through in vivo cellular approaches in Saccharomyces cerevisiae put their effort in my opinion into the CdS application in living organisms. The idea to use nanomaterials toward diagnostic or therapeutic treatment is important from a social point of view. Moreover, the ENMs application in biological experiments looks attractive too. Due to the above authors should strongly express the possibility of ENMs application in the introduction part. Moreover, in the whole article, I did not find any information about LD50 of CdS and toxicity of CdS in the experiment’s conditions. The toxicity of cadmium should be discussed and presented. Additionally, if authors discussed the mitochondria as a target it is obligated to show the level of discussed proteins (ELIS, WB, ect.). It is well known that mitochondria recruit proteins from the cytosol. Therefore, even though the article is well written and readable I cannot recommend it for publication in its current form.

Our proposal, as biotechnologists, is to offer a multidisciplinary and wide open approach to explain a biological phenomenon, that in this case has been consider in very specific context, as a proof of concept i) to introduce a potential, novel and reliable biophysical assay, which may offer a new perspective to analyze phenomena already described from biochemical and genetical point of view; ii) to add novel background information to the already studied pathways and mechanisms implicated in the nucleo-mitochondrial protein trafficking and how this impact with mitochondrial functionality.

For this reason the already characterize deletion mutant (delta)tom5, defective in import of mitochondrial proteins, has been taken in to account; and to give a further perspective to study the effect of a potential “next-gen” environmental stressor such as CdS QDs, which have been deeply studied in our lab, on different eukaryotic models (Marmiroli et al., 2016; Pasquali et al., 2017; Pagano et al., 2019; refs 5,7,41), in which all the information about mitochondrial target and expression relative levels have been fully characterized. Moreover, Gallo et al. (ref. 25), described in detail the effects of CdS QDs exposure from the proteomic point of view, utilizing different separation methods, such as 2D PAGE and iTRAQ.

In other words, we fully characterized the effects and mechanisms of response of yeast and the role of the mitochondrion in response to CdS QDs at genomic level (through KO mutant approach), transcriptomic level (microarray), proteomic level (2D PAGE and iTRAQ), and at physiological level (ROS production, glutathione, oxygen consumption, abundance of respiratory cytochromes; mitochondrial morphology). In this context, (delta)Tom5 mutant has been isolated from KO mutant screening as sensitive mutant [5], while at transcriptomic level as one of the nuclear gene with mitochondrial function with significant up-regulation [7,41]. The same (delta)Tom5 strain was also utilized for investigate defective mitochondrial organization [7].

On the technical point of view, FTIR can be used as alternative/complementary strategy to the robust biochemical approaches. It is important to consider that, at the present state, very few publications on the utilization of FTIR as technique to characterize cellular, but in particular mitochondrial response there are differential stressors.

For these reasons, keeping in mind the background information related to Tom5 biological role and the CdS QD treatment from previous papers, we decided to give a very specific context to this applied biology “proof of concept”: a characterization of the effects on different steps of the respiratory chain. The focus of the manuscript is strictly to understand how important the complementarity between the observations from the classical biochemical approach were and the more innovative FTIR (bio)physical approach, in gaining a reliable picture of the effects. 

Considering the interplay between nucleus and mitochondria, the problem of the cellular response of an environmental stress (the treatment), is mainly driven by the mitochondrial response at different levels: DNA replication, protein synthesis and assembly, protein trafficking, respiration, and oxidative phosphorylation. The use of genetics shown in the comparison between the two strains considered. The absence of a complete functional network of proteins in the mitochondrial membrane (as site of the respiratory chain) allowed for a more clear-cut of functional involved (in the effects) which goes well behind the production of ROS and the consequent oxidative stress.

In recently published paper (Paesano et al., 2023, ref 46) it has been shown that a cascade of genetic and epigenetic modulation drives the cell to mitophagy and autophagy or apoptosis, depending on the chemical form of the stressor (nano or ionic form).

Moreover, on the methodological point of view, FTIR spectroscopy was employed to extend the prior knowledge on this model system and to prove the applicability to the study of this “realistic system”. FTIR spectroscopy can provide both information on the chemical environment of the cells or organelles, and on their biomacromolecules. From the biological point of view, FTIR can be seen as a sort of vibrational phenotyping, assigning a specific vibrational pattern to the molecules constituting the sample. In this manuscript FTIR is used to characterize the molecular process involved in the different complexes of the respiratory chain, for both wt and (delta)tom5 in cells and in extracted mitochondria. Experimental data were then analyzed using principal component analysis in order to identify the infrared frequencies that vary the most during the treatment. The attribution of these signals to specific molecules was done according to the established literature.

To address the Rev3 queries, a series of changes have been performed on the manuscript:

The previous proteomic analysis performed on S. cerevisiae, treated with CdS QDs, has been highlighted in the introduction:

Lines 111-114: “The potential impairments at the level of protein trafficking from endoplasmic reticulum to the mitochondrion may influence different functions of the organelle itself, including the respiratory chain [23]. A details characterization of the effects of CdS QDs from the proteomic point of view, utilizing different separation methods, such as 2D PAGE and iTRAQ, has been carried out confirming the pivotal role of mitochondrion as main target of the CdS QD exposure [25].”

MIC has been included in the text (method section):

Lines 117-128: “The strain ypr133w-a (deltatom5) was identified as sensitive by mutant collection screening performed by Marmiroli et al. (2016) [5], in condition of treatment with CdS QDs, while (delta)tom5 molecular and functional characterization (ROS production, glutathione redox state, oxygen consumption, abundance of respiratory cytochromes; mitochondrial morphology), in condition of CdS QD exposure, were fully elucidated by Pasquali et al., 2017 [7]. The strains were grown in YPD (1% w/v Yeast extract, 2% w/v Peptone, 2% w/v Dextrose) liquid medium, with 0 mg L-1 or with 100 mg L-1 CdS QDs [7] sonicated Fisher Scientific Model 505 Sonic Dismembrator (Fisher Scientific, Waltham, MA), at 40% amplitude for 60−120s to maximize dispersion. Exposure tests have been conducted using sublethal conditions: a prior estimation of the CdS QDs minimal inhibitory concentration (MIC) was carried out (250 mg L−1), using concentrations ranging from 0 to 250 mg L−1 [5,7].”

Discussion has been improved:

Lines 364-412: “A cross-comparison of the two experiments highlighted potential concurrence or non-concurrence of the obtained data between FTIR, as direct measurement of the modification in molecular structure of the components implicated in the reactions, and the biochemical assay as an indirect measurement, which described at colorimetric variation due to the modification in the structure of a reagent triggered by mitochondrial environment: the two methods can be considered therefore complementary. The major aspects observed are correlated with i) the genetic background of the two S. cerevisiae strains, which are completely isogenic, with the exception of TOM5 gene deletion, and ii) with the CdS QDs treatment performed. This is consistent with the fact that, with the exception of treatment and genetic background of the strains utilized, all the other possible variables (medium, growth temperature, any other environmental condition) were maintained constant throughout the experiments.

[…]

The utilization of constitutively defective mitochondria, as those present in the (delta)tom5 mutant were functional to study effects, and mechanism developed by the yeast cell to deal with the interaction with CdS QDs, as fully characterized on genetic point of view through knock-out mutant screening [5], physiological point of view, in term of mitochondrial organization [7], and on functional point of view, at transcriptomic and proteomic level [25,41].”

Conclusions have been improved, including the nucleo-mitochondrial interplay.

Lines 423-436: “Utilization of yeast S. cerevisiae as a model system showed two main facilities: i) the possibility to grow yeast under controlled conditions, and ii) the flexibility of its life cycle which can occur on fermentable or non-fermentable carbon sources. This offers the unique opportunity to study the response in mutants whose mitochondrial functionality was compromised [4,5,7]. Knock-out mutants in specific mitochondrial function, as TOM5, are instrumental to test effects of artificial stressors, such as ENMs, when some of their potential targets are absent and to compare the results with the behaviour of the isogenic wild type strain [5]. The cellular response of an environmental stress (the treatment), which mainly is driven by the mitochondrial response at different levels: DNA replication, protein synthesis and assembly, protein trafficking, respiration, and oxidative phosphorylation [3,45]. The absence of a complete functional network of proteins in the mitochondrial membrane (as site of the respiratory chain) allowed for a more clear-cut of functional involved (in the effects) which goes well behind the production of ROS and the consequent oxidative stress.”

CdS QDs vs Cd ion discussion has been included from previous papers (in conclusions):

Lines 437-444: “Evidence obtained is relevant from three perspectives: i) from the methodological point of view, by comparing differential techniques, in order to magnify specific changes in biomolecules structure, organelle functionality, and cell viability; ii) from the genetic point of view, by comparing the behavior of differential yeast strains (wild type vs iso-genic deletion mutant), and iii) by CdS QDs treatment, as a major applied biology proof of concept of response to stress. It is important to state, as described in detail from Marmiroli at al., 2016 [5], that CdS QDs and Cd2+ exploit different mechanisms of response, highlighted by the very low overlap between genes and functions involved in the two pathways [5,41].”

Round 2

Reviewer 3 Report

I appreciated the author’s answer. However, I did not find any response to the crucial toxicity question for this study. Therefore before publication, the Cd toxicity must be discussed in the manuscript. In the present form, I do not recommend the article for publication.

Author Response

I appreciated the author’s answer. I did not find any response to the crucial toxicity question for this study. Therefore before publication, the toxicity must be discussed in the manuscript. In the present form I do not recommend the article for publication.  

In order to strengthen the comment previously included, we extended the discussion on the differences on CdS QDs vs Cd ion toxicity.

We have deeply studied this point in previous published papers, in different models systems, from yeast (simple eukaryote), plant model systems such as, Arabidopsis thaliana, and multiple human cell lines. All the references have been included in the manuscript (refs. 5,7,26,44,47).

To complete the elucidation of the mechanisms exploited in yeast, we also reported a new reference, specifically related to Cadmiun ion toxicity (Ruotyolo et al., 2008), new ref. 46, that was utilized for our previous studies as a comparison for the two Cd-based forms studied, in term of sensitive mutants isolated. All the crucial information was included in the new text.   

Lines 386-400: “Evidence obtained is relevant from three perspectives: i) from the methodological point of view, by comparing differential techniques, in order to magnify specific changes in biomolecules structure, organelle functionality, and cell viability; ii) from the genetic point of view, by comparing the behavior of differential yeast strains (wild type vs iso-genic deletion mutant), and iii) by CdS QDs treatment, as a major applied biology proof of concept of response to stress. It is important to state, as described in detail from Marmiroli at al., 2016 [5], that CdS QDs and Cd2+ exploit different mechanisms of response, highlighted by the very low overlap between genes and functions involved in the two pathways [5,41]: only 11 mutants on a panel of 114 isolated as CdS QDs sensitive mutants were commonly selected (9.6%), highlighting the differential mechanisms ex-ploited. This can be also testified by the differential MIC observed for CdS QDs (250 mg L-1) and CdSO4 (50 microM, which corresponds to 56 mg L-1) [46]. Moreover, the differential response mechanisms that interested Cd in nano and ionic forms, fully characterized at physico-chemical level by Pagano et al., 2022 [26], have been deeply elucidated in different model systems, from simple eukaryotes, to plants, and human cells [5,44,47].”

Round 3

Reviewer 3 Report

I strongly recommend inserting the toxicity of Cd in the introduction part before publication. At present, I cannot recommend this article for publication.

Author Response

I strongly recommend inserting the toxicity of Cd in the introduction part before publication. At present, I cannot recommend this article for publication.

In order to fulfill the Rev3 request we included more info about Cd toxicity in relationship with CdS QDs exposure in the introduction, keeping also the paragraphs included in the conclusions, with some edits.

Although, we have to remind that the main focus of the manuscript is not on the differences between the toxicity of Cd-based forms (nano or ionic) since this point has been fully elucidated in previously published papers. The main focus of the manuscript is the characterization of the effects on different steps of the respiratory chain as an applied biology “proof of concept”, utilizing different biotechnological (KO mutants defective in nucleo-mitochondrial trafficking vs wild type strain) and technical tools (colorimetric assay vs FTIR).

The introduction has been modified as reported.

Lines 92-105: “As an applied biology proof of concept, several targets from respiratory chains (Complex II, III, IV) were considered to assess the organellar functionality under stress condition (CdS QDs treatment) or in the absence of some mitochondrial function, through gene deletion, in a specific QD target, such as Tom5, as demonstrated by Marmiroli et al, 2016 [5]. Interestingly, by comparing the CdS QDs and Cd ion exposure, it has not been shown a marked overlap between commonly isolated sensitive mutants [5,23]: only 11 mutants on a panel of 114 isolated as CdS QD sensitive mutants were commonly observed as sensitive Cd2+ to exposure (9.6%), highlighting the differential response mechanisms exploited. Particular emphasis has been given in the case of CdS QDs response to abiotic stress and mitochondrial organization, while Cd2+ toxicity response was mainly related to metal detoxification, and vacuole transport [5,23]. Differences in the CdS QD and Cd2+ responses can be also testified by the differential MIC observed for CdS QDs (250 mg L-1) and CdSO4 (50 microM, which corresponds to 56 mg L-1 of Cd2+)[23].”

Conclusions have also been modified as reported.

Lines 399-423: “Evidence obtained is relevant from three perspectives: i) from the methodological point of view, by comparing differential techniques, in order to magnify specific changes in biomolecules structure, organelle functionality, and cell viability; ii) from the genetic point of view, by comparing the behavior of differential yeast strains (wild type vs iso-genic deletion mutant in term of nucleo-mitochondrial protein trafficking), and iii) by CdS QDs treatment, as a major applied biology proof of concept of response to stress. It is important to remind, as described in detail by Marmiroli et al., 2016 [5], that CdS QDs and Cd2+ exploit different mechanisms of response, highlighted by the very low overlap between genes and functions involved in the two pathways [5,42]. Moreover, the differential response mechanisms that interested Cd in nano and ionic forms, fully characterized at the physicochemical level by Pagano et al., 2022 [27], have been deeply elucidated in different model systems, from simple eukaryotes to plants, and human cells [5,27,45,47].

In the panel of potential methodologies which can be applied to the comprehension of in vivo mechanisms of homeostasis and cellular response, FTIR is a relatively new entry [48,49]. This technique can be successfully utilized to study in details the variation in the biomolecular structure of reagents and products during any biochemical reaction. The possibility of successfully applying this approach to in vivo study is instrumental to understanding the biological interactions, and relative interactors implicated in basic molecular pathways investigated. On the other hand, considering nanomaterials, as any other potential stressor, FTIR can be applied to the functional (direct or indirect) investigation of the physicochemical forms present in the cells, organs, or tissues, with particular regard to ENM in vivo biotransformation, as an essential step to understand the differences between the particle behavior, as compared to ionic counterparts, in complex systems (e.g. living organisms), as was already proved in other eukaryotic model systems [45,50].”

A new reference 50 has been included, with particular regard to nanomaterial biotransformation, which is functional to fulfill the comprehension of physicochemical parameters important to explain the differences between nano and ionic forms.